# Epidemiology of soil-transmitted helminthiasis among school-aged children in pastoralist communities of Kenya: A cross-sectional study

**Richelle W. Kihoro**[1]*, **Damaris Mulewa**[1,2], **Collins Okoyo**[1], **Dominic Ayaa**[2], **Agnes Korir**[2], **Doris W. Njomo**[1], **Charles Mwandawiro**[1], **Janet Masaku**[1]

**1** Eastern and Southern Africa Centre of International Parasite Control (ESACIPAC), Kenya Medical Research Institute (KEMRI), Nairobi, Kenya, **2** Department of Developmental Studies, Daystar University, Nairobi, Kenya

* richellewrk@gmail.com

## Abstract

### Introduction

Soil-transmitted helminthiasis (STH) are a major public health problem in Sub-Saharan Africa. In Kenya, the National School Based Deworming Program (NSBDP) was launched in 2012 with a goal of reducing STH prevalence in school-aged children (SAC) to <1%, however monitoring and evaluation results have consistently showed > 20% prevalence in Narok County. We conducted a study to investigate factors associated with STH infections among SAC in Narok County.

### Methodology

A cross-sectional study was conducted among 514 SAC from five schools in Trans Mara West sub-county, Narok County. The sub-county was selected because it had participating schools within the NSBDP with a high prevalence of STH infection. Participants were selected using systematic random sampling. Stool samples collected from participants were examined for STH eggs using Kato-Katz technique. An open data kit questionnaire was used to collect socio-demographics, household, and STH knowledge information from 139 of the 514 SAC. Descriptive statistics was used to summarize the data, prevalence and mean intensity of infections were calculated, and logistic regression used to determine factors associated with STH infections.

### Results

The overall prevalence of any STH infection was 24.6% (95%CI: 21.1–28.6). *Trichuris trichiura* 14.4% (95%CI: 11.7–17.8), *Ascaris lumbricoides* 12.5% (95%CI: 9.9–15.7) and hookworm 0%. From multivariable analysis the only factors significantly associated with increased risk of STH infection were, children attending Karda and Nkarano schools with aOR = 5.29 (95%CI: 1.45–19.24); p = 0.011 and aOR = 4.53 (95%CI: 1.29–15.97); p =

**Data Availability Statement:** The data set used for this manuscript is part of a bigger project that is currently ongoing. Access to the anonymized and

de-identified data set can be obtained by emailing the Kenya Medical Research Institute (KEMRI) Scientific Ethics Review Unit (SERU) at seru@kemri.go.ke this can ensure long term stability and availability of the data.

**Funding:** The author(s) received no specific funding for this work.

**Competing interests:** The authors have declared that no competing interests exist

**Abbreviations:** CI, Confidence Interval; ESACIPAC, Eastern and Southern Africa Centre of International Parasite Control; EPHS, Elimination as a Public Health Problem; KEMRI, Kenya Medical Research Institute; M & E, Monitoring & Evaluation; MDA, Mass Drug Administration; MoE, Ministry of Education; MoH, Ministry of Health; NACOSTI, National Commission for Science, Technology and Innovation; NSBDP, National School Based Deworming Program; ODK, Open Data Kit; OR, Odds Ratio; PC, Preventative Chemotherapy; SAC, School-aged Children; SBD, School-Based Deworming; SD, Standard Deviation; SERU, Scientific Ethics Review Unit; SSA, Sub-Saharan Africa; STH, Soil Transmitted Helminths; WHO, World Health Organization.

0.019 respectively. For *A. lumbricoides*, children attending Nkarano School were associated with a significant risk of infection with aOR = 7.81 (95%CI: 1.81–33.63); p = 0.006.

## Conclusions

Despite the ongoing work of NSBDP, the STH prevalence is still ≥ 20% in Trans Mara West sub-county Narok County, among SAC. This underscores the need for continued annual MDA. Additionally, if possible, drug combinations may effectively manage *T. trichiura*, the region's most common helminth. The study found a correlation between children attending specific schools and STH infection risk, suggesting the importance of health education and improved water, sanitation, and hygiene practices holistically both in schools and associated catchment areas that can act as STH reservoirs to alleviate the burden of STH.

## Introduction

Infections in humans with soil-transmitted helminthiasis (STHs) are most commonly caused by five species that is *Ascaris lumbricoides* the roundworm, *Trichuris trichiura* the whipworm, the hookworms *Necator americanus* and *Ancylostoma duodenale* and the threadworm *Strongyloides stercoralis*. These parasitic worms are among the twenty Neglected Tropical Diseases (NTDs) that affect the world's poorest populations [1]. They are more common among deprived communities in low- and middle-income countries that have inadequate access to water, sanitation and hygiene [2]. These parasitic disease infections have been found to be endemic in 166 countries worldwide [3]. Globally it is estimated that around 1.45 billion people are infected with STHs, with roundworms carrying the highest burden of over 800 million infections, followed by whipworm with over 450 infections and hookworm with over 400 million infections [3]. The ingestion of mature eggs is the route through which individuals acquire *T. trichiura* and *A. lumbricoides* infections while hookworm infections are mostly percutaneous [4]. Based on the routes of transmission children tend to be most infected as they have frequent contact with contaminated environments resulting to significant educational and nutritional effects once infected [5,6].

Sub-Saharan Africa (SSA) bears the highest burden of STHs globally in terms of prevalence and infection at 29% [7]. The regional progress to elimination of STHs in SSA shows a steady decline from 35% (or 74.6 million cases) in 2000 to 11% (or 33.9 million cases) in 2018 [8]. It is estimated that more than 5 million children in Kenya are at risk for contracting STHs and schistosomiasis infections [9]. The World Health Organization (WHO) recommends three main strategies for the control of STHs (a) preventive chemotherapy (PC) of high-risk groups such as school age children (SAC) and other vulnerable populations, (b) health education and (c) water, sanitation and hygiene interventions aimed at reducing contamination from soil [10–12].

The National School Based Deworming Program (NSBDP) was launched in 2012 through the Government of Kenya's, Ministry of Education (MoE) and Ministry of Health (MoH). The NSBDP was rolled out in 27 counties endemic for STHs and schistosomes that is Western, Nyanza, Rift Valley and Coast regions of the country. The aim of the NSBDP is to reduce the prevalence of infections in SAC to <1%, effectively eliminating them as a public health problem [13]. A comprehensive monitoring and evaluation (M & E) process measured the impact of the NSBDP on reducing the burden of STH and schistosomes through repeat cross-sectional surveys pre and post mass drug administration (MDA). The results of these have been

published for baseline [14], first three years [15], after five years [13] and year 6 (the year 2018) results [16] and more recently after this study year 10 results [17]. Generally, at baseline Narok and Vihiga counties which fall in the Rift Valley and Western regions respectively of Kenya had a prevalence of above 50% for any STH infection [14]. After three years of the program Western region was identified as a hotspot for STH infections [15]. After five rounds of MDA Western and Rift Valley regions with prevalence rates of 20.9% and 21.8% respectively remained the only two regions with a prevalence of greater than 20% [16]. These studies have consistently highlighted Narok County with a prevalence of over 20% for any STHs infection at baseline and after several rounds of MDA.

School-based deworming (SBD), notwithstanding, does not prevent rapid re-infection after treatment. For achieving a long- standing reduction in the prevalence of STHs, priority should be given to improving access to better sanitation in addition to providing PC and health education [18]. More often than not access to water, sanitation and hygiene is linked to socio-economic status. Moreover, factors associated with high prevalence of STHs among SAC have been documented [16,19,20]. Currently there is limited data on the prevalence of STHs and the associated risk factors in pastoral communities. Most of the existing research has focused on rice growing communities, where farmers and their children are highly susceptible to schistosomiasis infection as they stand for long hours in contaminated water and soil, additionally they are also susceptible to STHs infections as the routes of transmission are almost similar [20–23]. The dominant ethnic group in Narok County is the Maasai who are pastoralists. Compared to neighboring ethnic groups, Maasai individuals who rely on livestock herding as their primary source of income encounter notable challenges, particularly when it comes to the health of their children [24]. The socio-cultural issues and behavioral aspects related to water, sanitation, and hygiene, such as avoiding sharing pit latrines within families and their pastoral lifestyle, increase their vulnerability to STHs exposure [25].

In light of this information and the consistently high prevalence of STHs after multiple rounds of MDA, Narok county was chosen as the study area to understand factors associated with the high prevalence of STHs. Therefore, the paper presents findings of the prevalence and intensity of STHs and associated factors among SAC in Trans Mara West sub-county, Narok County, Kenya.

## Methods

### Study design and study site

This cross-sectional study was conducted from 29th January to 17th February 2018 in Trans Mara West sub-county, Narok county, Kenya. Narok County has six sub-counties namely Narok East, Narok North, Narok South, Narok West, Trans Mara East and Trans Mara West [26]. Narok County is situated in the southern part of the Rift Valley with geographical coordinates of 0˚ 50′ and 1˚ 50′ South and 35˚ 28′ and 36˚ 25′ East. Narok experiences two distinct rainy seasons every year, with an average rainfall ranging between 500 to 1800 mm per year. Soils here are sandy loam with high contents of the entire major plant nutrients such as calcium and magnesium. Trans Mara West sub-county was purposively selected based on the low reduction rates of STHs and schistosomiasis prevalence compared to other counties after five years of MDA [13]. Table 1 below shows the number of rounds of MDA provided at sub-county level and the reported coverage

### Study population

The target population comprised SAC (grade four to six, ranging from 4-16years) from five primary schools in Narok County. The five primary schools were Karda, Nkarano, Olereko,

**Table 1. STHs MDA and treatment coverage over time.**

| Year | Treatment Status | Coverage |
|---|---|---|
| 2013 | ✓ | 84% |
| 2014 | ✓ | 89% |
| 2015 | ✓ | 88% |
| 2016 | ✗ | - |
| 2017 | ✓ | 83% |

✓ Indicates Treated.

✗ Indicates Non- Treated.

Treatment coverage was reported at the sub-county level and all schools in the sub-county were reportedly treated.

Olookwaya and Pusanki. These schools were purposively selected based on the highest prevalence of STH among the schools monitored by the NSBDP, considering the data from Year 5 (the year 2017) prior to the study [13].

## Sample size determination, and sampling procedure

A sample size of 100 pupils per school was calculated to be adequate to detect a 5% change in prevalence of STH infections, assuming power of 80% and test size of 5% and considering anticipated variance in prevalence. The sampling frame encompassed students from grade four to six (formerly class four to six) attending the five primary schools participating in the study. Using the systematic random sampling technique, an equal interval was obtained depending on the number of children per class and the first child in each class was randomly selected. This sampling scheme and procedure is in line with a study conducted by the NSBDP [16].

## Ethical approval and consent to participate

The study was reviewed and approved by the Kenya Medical Research Institute (KEMRI), Scientific and Ethics Review Unit (SERU) 3558 and Daystar University Ethics Review Board under number 00301. The study also obtained a research license from the National Commission for Science, Technology and Innovation (NACOSTI) number 1056.

Before collecting data, the researchers visited the participating schools to provide an explanation of the study's purpose to the administration and leadership, and subsequently sought permission to conduct the study. Parents/ guardians of the children enrolled in the study were invited for meetings to discuss the aims, procedures, risks and benefits of participating in the study. Subsequently, written informed consent was obtained from parents/ guardians of the sampled participants. Study participants from grade six were also capable of providing assent. Each participant in the study was asked to provide a stool sample for parasitological examination of STH infection.

## Parasitological examination

The presence or absence of STH ova in stool samples was determined using the Kato-Katz technique [27,28]. Briefly, duplicate thick smears were made from each stool sample gathered using a sieve and a template calibrated to collect 41.7 mg of stool and covered with pre-soaked cellophane strips in glycerol-malachite green solution. Within 30 minutes, the slides were examined under a microscope at x100 and the counts were expressed as eggs per grams (epg) by multiplying with a standard conversion factor of 24. All parasitological examinations were carried out by trained medical laboratory technologists from KEMRI. A random examination

of 10% of the slides were re-read by senior technologists for quality assurance purpose. Children who tested positive received albendazole (400mg) treatment for STHs following the guidelines of the MoH and the WHO [29].

## Statistical analysis

The software application open data kit (ODK) was used to collect and enter the data into smartphones, which was then downloaded to a Microsoft Excel Spreadsheet, cleaned and exported to STATA version 15.1 (STATA Corporation, College Station, TX, USA) for analysis. Descriptive statistics such as proportions were used to summarize categorical variables while means were calculated for continuous variables. Prevalence and mean intensity for the infections were calculated and the 95% confidence intervals (CIs) determined using binomial regression and negative binomial regression models respectively. Infection intensities were classified into light, moderate and heavy infections according to WHO guidelines [11] and the prevalence of light, moderate and heavy infections together with 95% CIs obtained using binomial exact test. Questionnaires were used to collect data on socio-demographics, household and knowledge of STH factors. The questionnaire was administered to a subset of the surveyed individuals to determine their level of risk of STH infection. Hence risk factor analysis was performed only on the 139 children. Univariable analysis was conducted to determine association between infection prevalence and socio-demographics, household and knowledge of STH factors reporting odds ratios (ORs) and the 95%CIs while taking into account clustering by schools. The threshold for statistical significance was set at α = 0.05 and the significant variables were selected for inclusion in the multivariable model. For multivariable analysis, variables selected were included in the model in a sequential (block-wise) variable selection method. Adjusted ORs (aORs) and 95%CIs, of the best model, were obtained by mutually adjusting all the minimum generated variables using multivariable logistic regression model.

## Results

### Socio-demographic characteristics of the study population

A total of 514 school children participated from five schools namely; Karda 108 (20.9%), Nkarano 107 (20.7%), Olereko 108 (20.9%), Olookwaya 86 (16.7%) and Pusanki 105 (20.4%). Information on age and gender was obtained from 491 (95.5%) of the participants. The mean age of the children was 9.8 years (Standard deviation (SD):2.5, range: 4–16 years). Majority of the children, 292 (59.5%) were aged between 10 and 14 years, 188 children (38.3%) were less than 10 years and 11 (2.2%) of them were above 10 years. There were slightly more female children than male, 253 (51.5%) and 238 (48.5%) respectively.

### Prevalence of the infections

The overall prevalence of any STH infection was 24.6% (95%CI: 21.1–28.6) with species specific prevalence of 14.4% (95%CI: 11.7–17.8) for *T. trichiura* and 12.5% (95%CI: 9.9–15.7) for *A. lumbricoides* infection (Table 2). There were however no observed infections for hookworm.

 At school level, prevalence of any STH infection was highest in Olookwaya school at 33.7% (95%CI: 25.1–45.4), followed by Karda at 32.7% (95%CI: 24.9–42.9), Nkarano at 28.0% (95% CI: 20.7–37.9), Pusanki at 18.1% (95%CI: 12.0–27.2) and Olereko at 12.0% (95%CI: 7.2–20.0) (Fig 1). In terms of age categories, any STH prevalence was highest in children aged below 10 years at 31.9% (95%CI: 25.9–39.3), followed by those aged between 10 and 14 years, 20.3% (95%CI: 16.1–25.5) and least in those above 14 years, 9.1% (95%CI: 1.4–58.9). In terms of

**Table 2. Prevalence of STH infections among surveyed school children in Narok County.**

| Factors | No. of children sampled (%) | Prevalence %(95%CI); n | | |
|---|---|---|---|---|
| | | Any STH | T. trichiura | A. lumbricoides |
| Overall | 514 (100.0%) | 24.6% (21.1–28.6); n = 126 | 14.4% (11.7–17.8); n = 74 | 12.5% (9.9–15.7); n = 64 |
| **School** | | | | |
| Karda | 108 (21.0%) | 32.7% (24.9–42.9); n = 35 | 21.5% (14.9–30.9); n = 23 | 14.0% (8.8–22.4); n = 15 |
| Nkarano | 107 (20.8%) | 28.0% (20.7–37.9); n = 30 | 2.8% (0.9–8.6); n = 3 | 28.0% (20.7–37.9); n = 30 |
| Olereko | 108 (21.0%) | 12.0% (7.2–20.0); n = 13 | 9.3% (5.1–16.7); n = 10 | 4.6% (1.9–10.9); n = 5 |
| Olookwaya | 86 (16.7%) | 33.7% (25.1–45.4); n = 29 | 32.6% (24.0–44.1); n = 28 | 2.3% (0.6–9.1); n = 2 |
| Pusanki | 105 (20.4%) | 18.1% (12.0–27.2); n = 19 | 9.5% (5.3–17.2); n = 10 | 11.4% (6.7–19.5); n = 12 |
| **Age categories (years)** | | | | |
| <10 | 188 (38.3%) | 31.9% (25.9–39.3); n = 60 | 18.1% (13.3–24.5); n = 34 | 17.0% (12.4–23.3); n = 32 |
| 10–14 | 292 (59.5%) | 20.3% (16.1–25.5); n = 59 | 11.7% (8.5–16.0); n = 34 | 10.7% (7.6–14.9); n = 31 |
| >14 | 11 (2.2%) | 9.1% (1.4–58.9); n = 1 | 0 | 9.1% (1.4–58.9); n = 1 |
| **Gender** | | | | |
| Female | 253 (51.5%) | 25.3% (20.5–31.3); n = 64 | 15.4% (11.6–20.6); n = 39 | 12.6% (9.1–17.5); n = 32 |
| Male | 238 (48.5%) | 23.6% (18.8–29.7); n = 56 | 12.2% (8.7–17.2); n = 29 | 13.5% (9.8–18.6); n = 32 |

**n**; Indicates number of individuals tested positive for the infections.

gender, female children had the highest prevalence of any STH infection compared to males, 25.3% (95%CI: 20.5–31.3) and 23.6% (95%CI: 18.8–29.7) respectively (Table 2) (Fig 2).

For *T. trichiura*, prevalence was highest in Olookwaya School at 32.6% (95%CI: 24.0–44.1), followed by Karda at 21.5% (95%CI: 14.9–30.9), Pusanki at 9.5% (5.3–17.2), Olereko at 9.3%

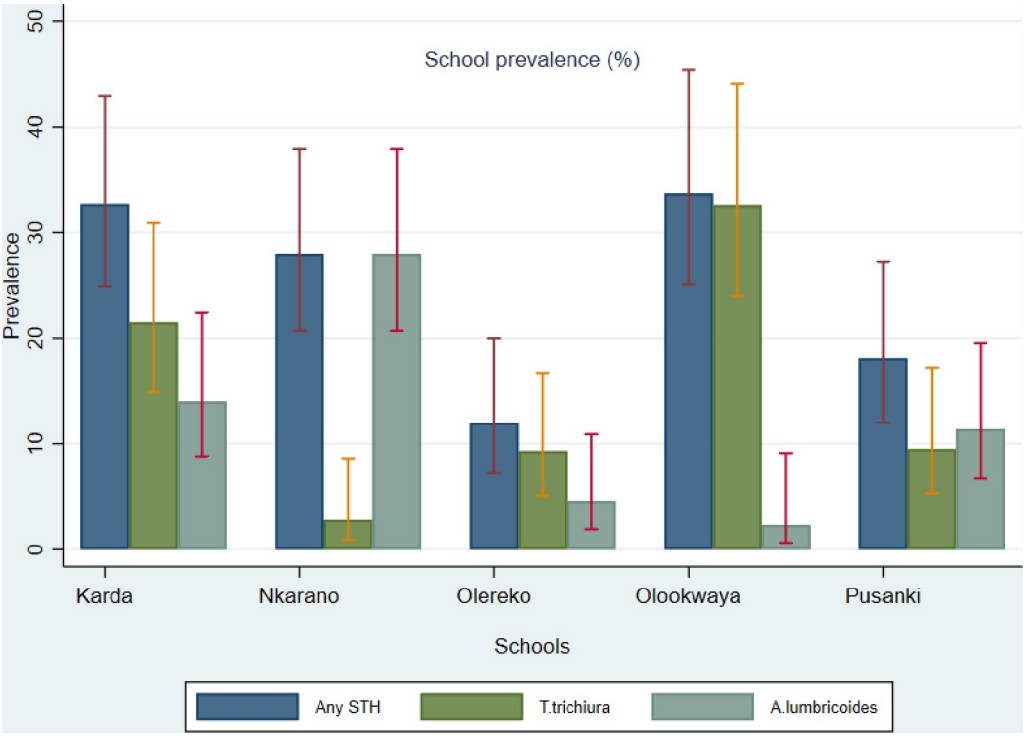

**Fig 1. School prevalence for different STH infection.**

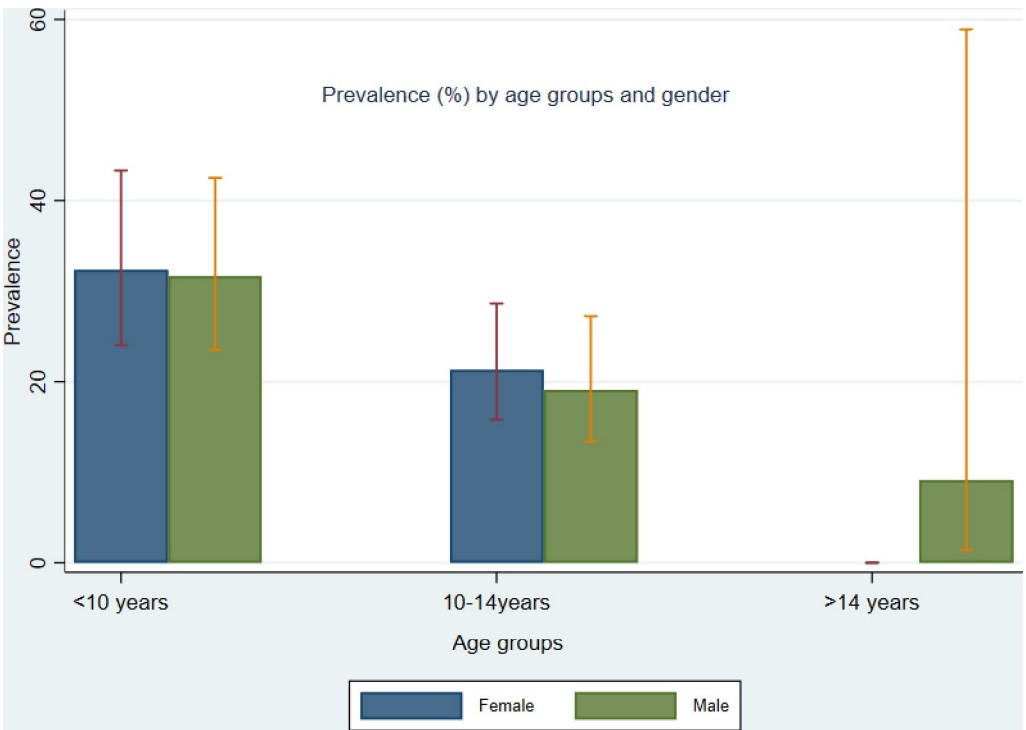

**Fig 2. Any STH prevalence by age groups and gender.**

(95%CI: 5.1–16.7) and least in Nkarano at 2.8% (95%CI: 0.9–8.6) (Fig 1). Infection with *T. trichiura* was highest in children aged below 10 years, 18.1% (95%CI: 13.3–24.5) followed by those aged between 10 and 14 years, 11.7% (95%CI: 8.5–16.0), however no infections were recorded among those aged above 14 years. Female children had the highest prevalence of *T. trichiura* followed by males, 15.4% (95%CI: 11.6–20.6) and 12.2% (95%CI: 8.7–17.2) respectively (Table 2).

Prevalence of *A. lumbricoides* was highest in Nkarano at 28.0% (95%CI: 20.7–37.9), followed by Karda at 14.0% (95%CI: 8.8–22.4), Pusanki at 11.4% (95%CI: 6.7–19.5), Olereko at 4.6% (95%CI: 1.9–10.9) and least in Olookwaya at 2.3% (95%CI: 0.6–9.1) (Fig 1). In terms of age categories, children below 10 years had the highest *A. lumbriocoides* infections, 17.0% (95%CI: 12.4–23.3) while those aged above 14 years had the lowest prevalence, 9.1% (95% CI:1.4–58.9). In addition, prevalence of *A. lumbricoides* was slightly higher in male children 13.5% (95%CI: 9.8–18.6) than females 12.6% (95%CI: 9.1–17.5) (Table 2).

## Mean intensity of the infections

The mean intensity of any STH infection was 870 epg (95%CI: 520–1454) with species specific intensity of 792 epg (95%CI: 363–1725) for *A. lumbricoides* and 78 epg (95%CI: 42–145) for *T. trichiura*. There were however no observed infections for hookworm (Table 3).

For any STH infection, mean intensity was highest in Nkarano School at 2461 epg (95%CI: 845–7169), followed by Karda at 874 epg (95%CI: 346–2206), Pusanki at 534 epg (95%CI: 139–2054), Olookwaya at 208 epg (95%CI: 84–517) and least in Olereko at 142 epg (95%CI: 29–692). In terms of age categories, mean intensity was highest in children aged below 10 years at 1407 epg (95%CI: 681–2907), followed by those aged between 10 and 14 years at 607 epg (95% CI: 284–1299) and least among children aged above 14 years at 302 epg (95%CI: 1–130309).

**Table 3. Mean intensity of infections among surveyed school children in Narok County.**

| Factors | No. of children sampled (%) | Mean intensity [epg(95%CI)] | | |
|---|---|---|---|---|
| | | Any STH | *Trichuris trichiura* | *Ascaris lumbricoides* |
| Overall | 514 (100.0%) | 870 (520–1454) | 78 (42–145) | 792 (363–1725) |
| Schools | | | | |
| Karda | 108 (21.0%) | 874 (346–2206) | 158 (52–478) | 716 (147–3490) |
| Nkarano | 107 (20.8%) | 2461 (845–7169) | 1 (0–13) | 2460 (845–7166) |
| Olereko | 108 (21.0%) | 142 (29–692) | 13 (3–61) | 129 (8–2036) |
| Olookwaya | 86 (16.7%) | 208 (84–517) | 193 (76–491) | 15 (0–828) |
| Pusanki | 105 (20.4%) | 534 (139–2054) | 47 (8–263) | 487 (83–2868) |
| Age categories (years) | | | | |
| <10 | 188 (38.3%) | 1407 (681–2907) | 152 (60–387) | 1255 (423–3722) |
| 10–14 | 292 (59.5%) | 607 (284–1299) | 34 (14–83) | 573 (187–1756) |
| >14 | 11 (2.2%) | 302 (1–130309) | - | 302 (1–130309) |
| Gender | | | | |
| Female | 253 (51.5%) | 716 (352–1457) | 69 (30–159) | 647 (217–1928) |
| Male | 238 (48.5%) | 1112 (508–2436) | 90 (32–251) | 1023 (339–3083) |

Male children had the highest mean intensity followed by females at 1112 epg (95%CI: 508–2436) and 716 epg (95%CI: 352–1457) respectively (Table 3).

For *T. trichiura*, the mean intensity was highest in Olookwaya School at 193 epg (95%CI: 76–491), followed by Karda at 158 epg (95%CI: 52–478), Pusanki at 47 epg (95%CI: 8–263), Olereko at 13 epg (95%CI: 3–61) and least in Nkarano at 1 epg (95%CI: 0–13). In terms of age categories, mean intensity was highest in children aged below 10 years at 152 epg (95%CI: 60–387), followed by those aged between 10 and 14 years at 34 epg (95%CI: 14–83) however, no infections were recorded in children aged above 14 years. Male children had the highest mean intensity followed by females at 90 epg (95%CI: 32–251) and 69 epg (95%CI: 30–159) respectively (Table 3).

For *A. lumbricoides*, the mean intensity was highest in Nkarano School at 2460 epg (95%CI: 845–7166), followed by Karda at 716 epg (95%CI: 147–3490), Pusanki at 487 epg (95%CI: 83–2868), Olereko at 129 epg (95%CI: 8–2036) and least in Olookwaya at 15 epg (95%CI: 0–828). In terms of age categories, mean intensity was highest in children aged below 10 years at 1255 epg (95%CI: 423–3722), followed by those aged between 10 and 14 years at 573 epg (95%CI: 187–1756) and least among children aged above 14 years at 302 epg (95%CI: 1–130309). Male children had the highest mean intensity compared to females at 1023 epg (95%CI: 339–3083) and 647 epg (95%CI: 217–1928) respectively (Table 3).

## Prevalence of light, moderate and heavy intensity of infections

The prevalence of light infections were 8.8% (95% CI: 6.5–11.5) and 13.0% (95% CI: 10.2–16.3) and prevalence of moderate infections were 3.7% (95% CI: 2.2–5.7) and 1.4% (95% CI: 0.5–2.8) respectively for *A. lumbricoides* and *T. trichiura*. There were however no observed infections for hookworms and no heavy intensity infections for either *A. lumbricoides or T. trichiura* (Table 4).

The prevalence of light infections of *A. lumbricoides* was highest in Nkarano school at 16.8% (95% CI: 10.3–25.3). In terms of age categories, the prevalence was highest in children aged below 10 years at 10.6% (95% CI: 6.6–16.0). Female children had the highest prevalence of light infections of *A. lumbricoides* at 9.9% (95% CI: 6.5–14.2).

**Table 4. Prevalence % (95% CI) of light, moderate and heavy intensity of infections among surveyed school children in Narok County.**

| Factors | Total number of children examined | Light intensity[1] | | | | Moderate intensity[2] | | | |
|---|---|---|---|---|---|---|---|---|---|
| | | *A. lumbricoides* | | *T. trichiura* | | *A. lumbricoides* | | *T. trichiura* | |
| | | Total number of light infections | Prevalence | Total number of light infections | Prevalence | Total number of moderate infections | Prevalence | Total number of moderate infections | Prevalence |
| Overall | 514 | 45 | 8.8 (6.5–11.5) | 67 | 13.0 (10.2–16.3) | 19 | 3. 7 (2.2–5.7) | 7 | 1.4 (0.5–2.8) |
| **School** | | | | | | | | | |
| Karda | 108 | 11 | 10.2 (5.2–17.5) | 21 | 19.4 (12.5–28.2) | 4 | 3.7 (1.0–9.2) | 2 | 1.8 (0.2–6.5) |
| Nkarano | 107 | 18 | 16.8 (10.3–25.3) | 3 | 2.8 (0.6–8.0) | 12 | 11.2 (5.9–18.8) | 0 | 0 |
| Olereko | 108 | 4 | 3.7 (1.0–9.2) | 10 | 9.3 (4.5–16.4) | 1 | 0.9 (0.0–5.1) | 0 | 0 |
| Pusanki | 86 | 10 | 11.6 (5.7–20.3) | 8 | 9.3 (4.1–17.5) | 2 | 2.3 (0.3–8.1) | 2 | 2.3 (0.3–8.1) |
| Olookwaya | 105 | 2 | 1.9 (0.2–6.7) | 25 | 23.8 (16.0–33.1) | 0 | 0 | 3 | 2.9 (0.6–8.1) |
| **Age category** | | | | | | | | | |
| <10 | 188 | 20 | 10.6 (6.6–16.0) | 29 | 15.4 (10.6–21.4) | 12 | 6.4 (3.3–10.8) | 5 | 2.7 (0.9–6.1) |
| 10–14 | 292 | 24 | 8.2 (5.3–12.0) | 32 | 11.0 (7.6–15.1) | 7 | 2.4 (1.0–4.9) | 2 | 0.7 (0.1–2.5) |
| >14 | 11 | 1 | 9.1 (0.3–41.3) | 0 | 0 | 0 | 0 | 0 | 0 |
| **Gender** | | | | | | | | | |
| Female | 253 | 25 | 9.9 (6.5–14.2) | 35 | 13.8 (9.8–18.7) | 7 | 2.8 (1.1–5.6) | 4 | 1. 6 (0.4–4.0) |
| Male | 238 | 20 | 8.4 (5.2–12.7) | 26 | 10.9 (7.3–15.6) | 12 | 5.0 (2.6–8.6) | 3 | 1.3 (0.3–3.6) |

[1]Light intensity 1–4,999 EPG for *A. lumbricoides*; 1–999 EPG for *T. trichiura*.

[2]Moderate intensity 5,000–49,999 EPG for *A. lumbricoides*; 1,000–9,999 EPG for *T. trichiura*.

The prevalence of light infections of *T. trichiura* was highest in Olookwayo school at 23.8% (95% CI: 16.0–33.1). In terms of age categories, the prevalence was highest in children aged below 10 years at 15.4% (95% CI:10.6–21.4). Female children had the highest prevalence of light infections of *T. trichiura* at 13.8% (95% CI:9.8–18.7).

The prevalence of moderate infections of *A. lumbricoides* was highest in Nkarano school at 11.2% (95% CI:5.9–18.8). In terms of age categories, the prevalence was highest in children aged below 10 years at 6.4% (95% CI:3.3–10.8). Male children had the highest prevalence of moderate infections of *A. lumbricoides* at 5.0% (95% CI:2.6–8.6).

The prevalence of moderate infections of *T. trichiura* was highest in Olookwayo school at 2.9% (95% CI:0.6–8.1). In terms of age categories, the prevalence was highest in children aged below 10 years at 2.7% (95% CI:0.9–6.1). Female children had the highest prevalence of moderate infections of *T. trichiura* at 1. 6% (95% CI:0.4–4.0).

## Household characteristics and knowledge of STHs

A questionnaire to collect data on household characteristics and knowledge of STH was administered to a sub-set of SAC (139); 29 from Nkarano School, 29 from Olereko, 29 from

Pusanki 28 from Karda School and 24 from Olookwaya. The reported average number of household size was 9 members (SD 3; range: 3–18) and majority 104 (74.8%) had 5 to 10 members. The overall average number of children in a household was 6 (SD 3; range: 1–20) and most households 95 (68.4%) had between 5 and 10 children. The average number of adults was 3 (SD 2, range: 1–22) and 127 households (91.4%) had less than 5 adults (Table 5).

Farming was the main economic activity in 82 households (58.9%). Most households, 109 (78.4%) used stream/river water for drinking and cooking. The main type of house construction material was mud 105 (75.5%). In term of home activities, most children, 58 (41.7%) were involved in fetching water, 50 (35.9%) were involved in herding, 34 (24.5%) in cooking and 11 of them (7.9%) in taking care of their siblings. Majority of the children, 123 (88.5%) reported always wearing shoes. A high proportion of households had a latrine facility, 107 (76.9%) and 98 children (89.9%) reported that they always used the latrine facility. Majority of the children reported that they mostly practice handwashing after visiting a latrine 112 (80.6%), before eating 72 (51.8%) or after eating 21 (15.1%) (Table 5).

Most children, 113 (81.3%), reported that they had prior knowledge about STHs, 54 (38.9%) citing ingestion of dirty food, 31 (22.3%) dirty water and 5 (3.6%) contact with infected water as transmission routes, however, 38 (27.3%) did not know. The most common symptom of STH reported was stomach ache 91 (65.5%). However, 28 (20.1%) did not know of any STH symptoms. Participants reported risk of contracting STH infections as anemia 6 (4.3%), skin rashes 8 (5.8%), internal bleeding or clotting 2 (1.4%), however, 119 (85.6%) did not know of any risks. Participants also reported that the causes of STH infections in children were; eating dirty food 49 (35.3%), poor sanitation 27 (19.4%) and poor personal hygiene 17 (12.2%). The ways to reduce STH infection stated were; deworming 97 (69.8%), health education 12 (8.6%), ensuring of a clean environment 17 (12.2%), and eating balanced diet 13 (9.4%) (Table 5).

## Factors associated with STH infections

**Univariable analysis.** From the univariable analysis, the following factors were associated with increased risk of any STH infections; children in Karda and Nkarano schools with OR = 4.80 (95%CI: 1.42–16.18); p = 0.011 and OR = 3.90 (95%CI: 1.16–13.08); p = 0.027 respectively, those whose parents were in employment OR = 2.31 (95%CI: 1.33–3.98); p = 0.003, and children from households using rainwater as the drinking water source OR = 2.02 (95%CI: 1.39–2.93); p<0.001. However, the factors associated with reduced risk of any STH infections were; children who reported that taking dirty water was a way of increasing transmission OR = 0.64 (95%CI: 0.26–1.57); p = 0.015, and those who reported that vomiting was a symptom of STH infection OR = 0.58 (95%CI: 0.40–0.84); p = 0.004 (Table 5).

For *T. trichiura* infection, factors associated with increased risk were; households having more than 10 children OR = 3.50 (95%CI: 1.47–8.32); p = 0.005, parents/guardians who were employed OR = 3.02 (95%CI: 1.38–6.61); p = 0.006, households using borehole/well as the drinking water source OR = 2.08 (95%CI: 1.04–4.19); p = 0.039, those who reported that stomachaches were a symptom of STH infection OR = 1.85 (95%CI: 1.15–2.96); p = 0.011 and those who reported that eating balanced diet was a way to reduce STH transmission OR = 3.31 (95% CI:1.33–8.25); p = 0.010. However, the factors associated with reduced risk of *T. trichiura* infection was observed among children engaged in cooking chores OR = 0.22 (95%CI: 0.05–0.99); p = 0.049, children who always wore shoes compared to those who occasionally wore shoe OR = 0.30 (95%CI: 0.09–0.95); p = 0.040 (Table 5).

For *A. lumbricoides*, factors associated with increased risk of infection were; children in Nkarano School with OR = 7.04 (95%CI: 1.73–28.59); p = 0.006, households having between 5

**Table 5. Univariable analysis of risk factors associated with infections among surveyed children.**

| Factors | No. of children sampled N = 139 | Univariable logistic [OR (95%CI); p-value] | | |
| --- | --- | --- | --- | --- |
| | | Any STH n = 46 | *T. trichiura* n = 25 | *A. lumbricoides* n = 27 |
| **Schools** | | | | |
| **Karda** | 28 (20.1%) | 4.80 (1.42–16.18); p = 0.011* | 2.08 (0.54–8.11); p = 0.290 | 3.47 (0.81–14.77); p = 0.093 |
| **Nkarano** | 29 (20.9%) | 3.90 (1.16–13.08); p = 0.027* | 0.46 (0.08–2.75); p = 0.397 | 7.04 (1.73–28.59); p = 0.006* |
| **Olereko** | 29 (20.9%) | 0.77 (0.18–3.21); p = 0.717 | 0.72 (0.15–3.55); p = 0.688 | 0.64 (0.09–4.16); p = 0.642 |
| **Olookwaya** | 24 (17.3%) | 3.43 (0.97–12.08); p = 0.055 | 3.75 (0.98–14.33); p = 0.053 | 0.38 (0.04–3.88); p = 0.412 |
| **Pusanki** | 29 (20.9%) | Reference | | |
| **Age categories (years)** | | | | |
| **<10** | 22 (16.2%) | Reference | | |
| **10–14** | 114 (83.8%) | 0.93 (0.38–2.26); p = 0.879 | 1.47 (0.59–3.68); p = 0.410 | 0.88 (0.36–2.18); p = 0.782 |
| **Household members** | | | | |
| **No. of members** | | | | |
| **<5** | 3 (2.2%) | Reference | | |
| **5–10** | 104 (74.8%) | 0.81 (0.07–9.30); p = 0.866 | 0.59 (0.25–1.37); p = 0.216 | 0.39 (0.07–2.24); p = 0.292 |
| **>10** | 32 (23.0%) | 1.76 (0.10–30.81); p = 0.697 | - | 0.78 (0.08–7.98); p = 0.836 |
| **No. of children** | | | | |
| **<5** | 32 (23.0%) | Reference | | |
| **5–10** | 95 (68.4%) | 0.84 (0.34–2.09); p = 0.706 | 1.53 (0.76–3.07); p = 0.237 | 0.56 (0.29–1.09); p = 0.090 |
| **>10** | 12 (8.6%) | 1.91 (0.19–18.66); p = 0.578 | 3.50 (1.47–8.32); p = 0.005 | 1.50 (0.12–18.28); p = 0.751 |
| **No. of adults** | | | | |
| **<5** | 127 (91.4) | Reference | | |
| **5–10** | 11 (7.9%) | 3.95 (0.82–19.05); p = 0.087 | 1.79 (0.49–6.47); p = 0.375 | 2.58 (1.32–5.06); p = 0.006 |
| **>10** | 1 (0.7%) | - | - | - |
| **Family Occupation** | | | | |
| **Farming** | | | | |
| **Yes** | 82 (58.9%) | 0.98 (0.43–2.26); p = 0.966 | 0.71 (0.28–1.77); p = 0.459 | 1.23 (0.58–2.62); p = 0.593 |
| **No** | 57 (41.0%) | Reference | | |
| **Livestock keeping** | | | | |
| **Yes** | 52 (37.4%) | 1.12 (0.70–1.77); p = 0.642 | 0.93 (0.49–1.74); p = 0.817 | 1.19 (0.62–2.28); p = 0.599 |
| **No** | 87 (62.6%) | Reference | | |
| **Business** | | | | |
| **Yes** | 41 (29.5%) | 0.66 (0.34–1.28); p = 0.217 | 0.92 (0.30–2.78); p = 0.875 | 0.80 (0.56–1.14); p = 0.222 |
| **No** | 98 (70.5%) | Reference | | |
| **Employed** | | | | |
| **Yes** | 20 (14.4%) | 2.31 (1.33–3.98); p = 0.003 | 3.02 (1.38–6.61); p = 0.006 | 1.47 (0.50–4.29); p = 0.482 |
| **No** | 119 (85.6%) | Reference | | |
| **Water sources for drinking and cooking** | | | | |
| **Piped/tap water** | | | | |
| **Yes** | 12 (8.6%) | 0.38 (0.07–1.96); p = 0.246 | 0.39 (0.03–4.42); p = 0.447 | 0.35 (0.08–1.51); p = 0.160 |
| **No** | 127 (91.4%) | Reference | | |
| **Borehole/well** | | | | |
| **Yes** | 10 (7.2%) | 2.15 (0.53–8.65); p = 0.283 | 2.08 (1.04–4.19); p = 0.039 | 1.88 (0.46–7.72); p = 0.384 |
| **No** | 129 (92.8%) | Reference | | |
| **Rainwater** | | | | |
| **Yes** | 19 (13.7%) | 2.02 (1.39–2.93); p<0.001 | 1.26 (0.59–2.70); p = 0.558 | 2.18 (1.33–3.57); p = 0.002 |
| **No** | 120 (86.3%) | Reference | | |

(*Continued*)

**Table 5.** (Continued)

| Factors | No. of children sampled N = 139 | Univariable logistic [OR (95%CI); p-value] | | |
|---|---|---|---|---|
| | | Any STH n = 46 | *T. trichiura* n = 25 | *A. lumbricoides* n = 27 |
| **Stream/river** | | | | |
| Yes | 109 (78.4%) | 0.82 (0.24–2.79); p = 0.747 | 0.84 (0.38–1.89); p = 0.681 | 0.95 (0.24–3.85); p = 0.948 |
| No | 30 (21.6%) | Reference | | |
| **Type of house material** | | | | |
| Mud | 105 (75.5%) | Reference | | |
| Cement | 28 (20.1%) | 0.77 (0.54–1.09); p = 0.139 | 1.32 (0.77–2.27); p = 0.319 | 0.67 (0.49–0.89); p = 0.007 |
| Wood | 3 (2.2%) | 0.96 (0.07–12.39); p = 0.974 | 2.42 (0.09–62.54); p = 0.595 | 2.00 (0.22–18.25); p = 0.539 |
| Others | 3 (2.2%) | 0.96 (0.49–1.85); p = 0.899 | - | 2.00 (0.74–5.44); p = 0.174 |
| **Type of activities involved with at home** | | | | |
| **Cooking** | | | | |
| Yes | 34 (24.5%) | 0.79 (0.49–1.29); p = 0.357 | 0.22 (0.05–0.99); p = 0.049 | 1.74 (0.66–4.57); p = 0.261 |
| No | 105 (75.5%) | Reference | | |
| **Fetching water** | | | | |
| Yes | 58 (41.7%) | 0.65 (0.24–1.74); p = 0.388 | 0.48 (0.12–1.87); p = 0.290 | 1.15 (0.56–2.35); p = 0.708 |
| No | 81 (58.3%) | Reference | | |
| **Herding** | | | | |
| Yes | 50 (35.9%) | 1.23 (0.74–2.04); p = 0.433 | 1.85 (0.81–4.19); p = 0.143 | 0.56 (0.29–1.09); p = 0.092 |
| No | 89 (64.0%) | Reference | | |
| **Taking care of siblings** | | | | |
| Yes | 11 (7.9%) | 0.74 (0.24–2.28); p = 0.601 | 1.01 (0.25–4.07); p = 0.984 | 0.39 (0.05–3.09); p = 0.374 |
| No | 128 (92.1%) | Reference | | |
| **Frequency of shoe wearing** | | | | |
| Always | 123 (88.5%) | 0.80 (0.24–2.70); p = 0.723 | 0.30 (0.09–0.95); p = 0.040 | 4.02 (0.50–32.07); p = 0.189 |
| Occasionally | 16 (11.5%) | Reference | | |
| **Latrine factors at home** | | | | |
| **Presence of toilet** | | | | |
| Yes | 107 (76.9%) | 1.29 (0.56–2.98); p = 0.554 | 1.76 (0.83–3.74); p = 0.139 | 0.95 (0.54–1.66); p = 0.845 |
| No | 32 (23.0%) | Reference | | |
| **Always used latrine facility** | | | | |
| Yes | 98 (89.9%) | 1.29 (0.67–2.48); p = 0.441 | 2.09 (0.21–21.04); p = 0.528 | 1.15 (0.33–4.00); p = 0.822 |
| No | 11 (10.1%) | Reference | | |
| **Handwashing times** | | | | |
| **Before eating** | | | | |
| Yes | 72 (51.8%) | 1.02 (0.50–2.07); p = 0.941 | 1.23 (0.50–3.01); p = 0.652 | 0.83 (0.42–1.67); p = 0.610 |
| No | 67 (48.2%) | Reference | | |
| **After eating** | | | | |
| Yes | 21 (15.1%) | 0.78 (0.28–2.16); p = 0.535 | 2.08 (0.79–5.53); p = 0.140 | 0.65 (0.29–1.44); p = 0.293 |
| No | 118 (84.9%) | Reference | | |
| **After visiting the latrine** | | | | |
| Yes | 112 (80.6%) | 0.81 (0.34–1.93); p = 0.568 | 0.96 (0.36–2.53); p = 0.929 | 1.08 (0.37–3.12); p = 0.893 |
| No | 27 (19.4%) | Reference | | |
| **Knowledge of STHs** | | | | |
| Yes | 113 (81.3%) | 0.92 (0.37–2.26); p = 0.768 | 1.85 (0.70–4.90); p = 0.214 | 0.76 (0.32–1.81); p = 0.538 |
| No | 26 (18.7%) | Reference | | |

(*Continued*)

**Table 5.** (Continued)

| Factors | No. of children sampled N = 139 | Univariable logistic [OR (95%CI); p-value] | | |
|---|---|---|---|---|
| | | **Any STH n = 46** | **T. trichiura n = 25** | **A. lumbricoides n = 27** |
| **Ways of STH transmission** | | | | |
| **Contact with infected water** | | | | |
| Yes | 5 (3.6%) | 3.17 (0.51–19.70); p = 0.232 | 7.64 (0.99–58.43); p = 0.050 | 2.91 (0.43–19.43); p = 0.271 |
| No | 134 (96.4%) | Reference | | |
| **Ingesting dirty food** | | | | |
| Yes | 54 (38.9%) | 1.17 (0.57–2.39); p = 0.653 | 1.58 (0.37–6.77); p = 0.536 | 1.10 (0.75–1.63); p = 0.622 |
| No | 85 (61.2%) | Reference | | |
| **Ingesting dirty water** | | | | |
| Yes | 31 (22.3%) | 0.64 (0.26–1.57); p = 0.015 | 0.85 (0.47–1.53); p = 0.579 | 0.99 (0.69–1.42); p = 0.975 |
| No | 108 (77.7%) | Reference | | |
| **Reported symptoms of STHs** | | | | |
| **Stomach ache** | | | | |
| Yes | 91 (65.5%) | 0.98 (0.47–2.07); p = 0.931 | 1.85 (1.15–2.96); p = 0.011 | 0.87 (0.74–1.02); p = 0.096 |
| No | 48 (34.5%) | Reference | | |
| **Head ache** | | | | |
| Yes | 4 (2.9%) | 2.12 (0.29–15.53); p = 0.657 | 1.53 (0.31–7.43); p = 0.600 | 1.45 (0.07–28.67); p = 0.806 |
| No | 134 (96.4%) | Reference | | |
| **Dizziness** | | | | |
| Yes | 4 (2.9%) | 2.07 (0.28–15.17); p = 0.257 | 1.54 (0.24–10.03); p = 0.650 | 4.40 (1.59–12.14); p = 0.004 |
| No | 135 (97.1%) | Reference | | |
| **Vomiting** | | | | |
| Yes | 13 (9.4%) | 0.58 (0.40–0.84); p = 0.004 | 0.35 (0.05–2.46); p = 0.294 | 1.28 (0.53–3.05); p = 0.585 |
| No | 126 (90.7%) | Reference | | |
| **Reported risks of STH infections** | | | | |
| **Anaemia** | | | | |
| Yes | 6 (4.3%) | 2.09 (0.41–10.80); p = 0.128 | 0.91 (0.12–6.74); p = 0.925 | 2.16 (0.79–5.85); p = 0.130 |
| No | 133 (95.7%) | Reference | | |
| **Skin rashes** | | | | |
| Yes | 8 (5.8%) | 0.66 (0.13–3.40); p = 0.530 | 1.57 (0.65–3.76); p = 0.316 | 0.58 (0.05–6.25); p = 0.651 |
| No | 131 (94.2%) | Reference | | |
| **Internal bleeding/clotting** | | | | |
| Yes | 2 (1.4%) | 2.04 (0.12–33.44); p = 0.697 | 4.71 (0.21–105.28); p = 0.328 | 4.27 (0.08–225.16); p = 0.473 |
| No | 137 (98.6%) | Reference | | |
| **Reported causes of STH infections in children** | | | | |
| **Poor sanitation** | | | | |
| Yes | 27 (19.4%) | 1.51 (0.64–3.59); p = 0.303 | 1.39 (0.68–2.89); p = 0.367 | 2.06 (0.77–5.54); p = 0.152 |
| No | 112 (80.6%) | Reference | | |
| **Poor personal hygiene** | | | | |
| Yes | 17 (12.2%) | 1.12 (0.39–3.24); p = 0.867 | 0.97 (0.23–4.13); p = 0.971 | 1.32 (0.41–4.31); p = 0.640 |
| No | 122 (87.8%) | Reference | | |
| **Eating dirty food** | | | | |
| Yes | 49 (35.3%) | 0.73 (0.34–1.54); p = 0.446 | 0.84 (0.31–2.27); p = 0.728 | 0.73 (0.29–1.78); p = 0.487 |
| No | 90 (64.8%) | Reference | | |
| **Reported ways to reduce STH infection** | | | | |

(*Continued*)

**Table 5.** (Continued)

| Factors | No. of children sampled N = 139 | Univariable logistic [OR (95%CI); p-value] | | |
|---|---|---|---|---|
| | | Any STH n = 46 | T. trichiura n = 25 | A. lumbricoides n = 27 |
| **Deworming** | | | | |
| Yes | 97 (69.8%) | 0.85 (0.39–1.81); p = 0.505 | 0.72 (0.45–1.18); p = 0.192 | 1.29 (0.34–4.96); p = 0.702 |
| No | 42 (30.2%) | Reference | | |
| **Health education** | | | | |
| Yes | 12 (8.6%) | 1.49 (0.45–5.01); p = 0.523 | 1.59 (0.46–5.48); p = 0.462 | 0.82 (0.43–1.53); p = 0.528 |
| No | 127 (91.4%) | Reference | | |
| **Clean environment** | | | | |
| Yes | 17 (12.2%) | 1.12 (0.39–3.24); p = 0.699 | 1.48 (0.41–5.35); p = 0.550 | 0.88 (0.35–2.22); p = 0.778 |
| No | 122 (87.8%) | Reference | | |
| **Eating good food** | | | | |
| Yes | 13 (9.4%) | 1.84 (0.58–5.84); p = 0.202 | 3.31 (1.33–8.25); p = 0.010 | 1.28 (0.50–3.23); p = 0.609 |
| No | 126 (90.7%) | Reference | | |

*significant variables at p<0.05.

**n;** Indicates number of individuals tested positive for the infections.

and 10 adults OR = 2.58 (95%CI: 1.32–5.06); p = 0.006, households using rainwater for drinking and cooking OR = 2.18 (95%CI:1.33–3.57); p = 0.002, those who reported dizziness as a symptom of STH OR = 4.40 (95%CI:1.59–12.14); p = 0.004. However, factors associated with reduced risk of *A. lumbricoides* infection were; children living in houses constructed using cement OR = 0.67 (0.49–0.89); p = 0.007 (Table 5).

**Multivariable analysis.** From the multivariable analysis, the factors significantly associated with increased risk of STH infections were; children from Karda and Nkarano schools with aOR = 5.29 (95%CI: 1.45–19.24); p = 0.011 and aOR = 4.53 (95%CI: 1.29–15.97); p = 0.019 respectively (Table 6). Although marginally non-significant, households using rainwater source for drinking had increased risk of STH infection aOR = 3.14 (95%CI: 0.97–10.17); p = 0.056. For *A. lumbricoides*, children in Nkarano School were associated with a significant risk of infection with aOR = 7.81 (95%CI: 1.81–33.63); p = 0.006 (S1 Table). For *T. trichiura* infection, none of the factors showed significant associations (S2 Table).

## Discussion

The NSBDP, impact has been evaluated in several studies [13–16]. This study offers an in-depth evaluation of the prevalence, intensity and associated risk factors of STH in selected schools participating in the NSBDP within Trans Mara West sub-county Narok county, Kenya. Narok has consistently exhibited a prevalence exceeding 20% for any STH infection since commencement of the NSBDP and up to this study [17]. The data generated from this study will contribute to the existing knowledge and aligns with the overarching objective of eliminating NTDs by 2030 [30].

According to the current treatment guidelines from the WHO, in this region where the prevalence is 24.6%, it is recommended to continue the previous MDA schedule, which involves administering treatments once a year [31]. These findings underscore that depending exclusively on MDA is inadequate to achieve STH elimination, given the persistently high STH burden in Narok County despite numerous MDA rounds. Studies indicate that

**Table 6. Multivariable analysis of risk factors associated with STH infections.**

| Factors | Adjusted Odds Ratio [aOR(95%CI)] | p-value |
|---|---|---|
| **School** | | |
| Karda | 5.29 (1.45–19.24) | 0.011* |
| Nkarano | 4.53 (1.29–15.97) | 0.019* |
| Olereko | 0.86 (0.19–3.83) | 0.840 |
| Olookwaya | 3.55 (0.93–13.55) | 0.064 |
| Pusanki | Reference | |
| **Family occupation** | | |
| Employed | | |
| Yes | 2.51 (0.88–7.19) | 0.086 |
| No | Reference | |
| **Water source for drinking and cooking** | | |
| Rainwater | | |
| Yes | 3.14 (0.97–10.17) | 0.056 |
| No | Reference | |
| **Ways of STH transmission** | | |
| Ingesting dirty water | | |
| Yes | 0.63 (0.22–1.82) | 0.393 |
| No | Reference | |
| **Reported symptoms of STHs** | | |
| Vomiting | | |
| Yes | 0.46 (0.10–2.12) | 0.319 |
| No | Reference | |

additional approaches such as enhancing access to clean water and better sanitation practices, as well as using more effective drugs, should be implemented to attain the goal of eradicating STH [32]. It is plausible that the results can be attributed to the high likelihood of individuals being re-infected even after receiving multiple rounds of treatment and studies have emphasized the importance of implementing other approaches to complement MDA [33].

The concept of elimination of STHs as a public health problem (EPHP) is defined as proportion of moderate and heavy intensity infections that are below <2% [30]. Overall, in our study area *T. trichiura* has met the criteria for EPHP at 1.4%, while *A. lumbricoides* at 3.7% has not yet achieved EPHP status. Furthermore, out of the five schools assessed for EPHH only two have achieved this threshold for *A. lumbricoides* while three schools have achieved this for *T. trichiura*. It has been shown that infection transmission and reduction of STHs can be influenced by environmental conditions, household WASH factors and population density factors [19].

With regards to species specific helminths, the most common species was *T. trichiura* (14.4%) followed by *A. lumbricoides* (12.5%). No infections were detected for hookworm. In Togo, a study revealed a prevalence of 11.1% for hookworms, along with negligible prevalence of 0.4% and 0.3% for *T. trichiura* and *A. lumbricoides*, respectively, potentially due to the frequent distribution of ivermectin for onchocerciasis treatment [34]. In our study, the prevalence of *T. trichiura* and *A. lumbricoides* was lower than that in a different study within the same region in 2019 [35]. Our findings are in line with the NSBDP conducted in the same year (2018), which reported an identical prevalence of 14.4% for *T. trichiura* and a comparable prevalence of 12.4% for *A. lumbricoides*. Moreover, the NSBDP study did not identify any cases of hookworm infections [16].

The higher prevalence of *T. trichiura* as compared to other helminths could be due to the reduced sensitivity to a single dose of albendazole, which is the current available medication employed for MDA [36]. This deduction concurs with modelling studies, which have demonstrated that when drug effectiveness is inadequate, STH infections will persist [37]. The prevalence of *A. lumbricoides* observed can be accounted for by the occurrence of re-infections, as *A. lumbricoides* has demonstrated higher re-infection rates compared to other STHs following MDA [13]. As majority of the students (88.5%) reported wearing shoes always this may have contributed to the lack of hookworm infections whose mode of transmission is penetrating the skin. It is also possible to hypothesize that optimal temperature and humidity conditions in the soil were not attained for hatching of the larval stage of hookworm for completion of the life cycle [38]. Findings in the present study are consistent with a previous study that showed increase in temperature lead to an increase in prevalence of *T. trichiura* and *A. lumbricoides* while an increase in temperature lead to a decrease in the prevalence of hookworms [39]. This may be the case in our study area as it is majorly an arid and semi-arid area with pastoralists communities. Research indicates higher prevalence of hookworms in adults rather than children, this both elucidates our findings and prompts concerns regarding the adequacy of the current treatment recommendations [40].

Whilst this study did not detect any hookworm species, a previous study reported a prevalence of 46.80% for *Ancylostoma caninum* in dog feces in Narok (Maasai Mara) [41]. South of Kenya in Morogoro, Tanzania all four known species of canine hookworms were identified from dog faeces [42]. Therefore because of the close interface between human, livestock and wild animals in Narok County control measures such as deworming of dogs and public health awareness of dogs as reservoirs of zoonotic hookworms should be implemented.

Generally, prevalence and intensity of STH and species-specific helminths increased with a decrease in age of the study participants. A study carried out in Nigeria reported a similar pattern of declining STH infections as age increases, which is consistent with the findings of this study [43]. A comparable trend was also identified in a study conducted in a different location [44]. The decreased prevalence of STH in the older age groups may be attributed to better hygiene practices and increased awareness regarding the spread of these infections. The higher prevalence of STH infections among younger children may be attributed to factors such as eating contaminated food from the ground and poor hygiene practices, such as having untrimmed and contaminated finger nails [45]. Younger children may have a higher burden of infection due to their recent transition from pre-school aged children (PSAC) to SAC, which are not covered as of this study by the NSBDP.

The analysis of STH infection risk factors revealed that SAC attending specific schools i.e. Karda and Nkarano schools had a significantly higher risk of STH infections. Additionally, presence of *A. lumbricoides* infection was significantly associated with children attending Nkarano school. The present study's results are consistent with a prior investigation that found a high prevalence of STH infection in Nkarano school [35]. Previous research conducted in Kenya identified a correlation between school sanitation factors such as type of toilet, toilet conditions and pupils to latrine ratio and the prevalence of overall or worm-specific infections [46]. Furthermore, in Kenyan schools, the recommended minimum requirement stipulates an ideal latrine to pupil ratio of 1:25 for girls and 1:30 for boys [47]. Although our study found only a marginally non-significant association between using rain water for drinking or cooking and STH infection, previous research has shown that consuming water that is mostly untreated increases risk of STH infection [48,49].

## Study limitations

Our study relied on single stool samples, disregarding the temporal variation in egg shedding associated with STH infections, where individuals may have varying egg output on different days. Secondly, the study's scope was restricted to Narok County which is primarily inhabited by pastoral communities residing in arid or semi-arid regions potentially reducing the applicability of the findings to other communities or regions. Additionally, household and knowledge risk factors of STH relied on self-reported data, which means that respondents might have provided responses based on socially acceptable norms rather than complete accuracy. Lastly, MDA coverage was reported at sub-county level rather than the school level, due to data availability constraints. Thus, some schools may have experienced low MDA coverage due to factors such as accessibility, reduced school attendance or compliance among teachers or children.

## Conclusions

The study revealed a 24.6% prevalence of any STH among SAC in Trans Mara West Sub-county, Narok County, Kenya, reaffirming STH as an ongoing public health concern. This emphasizes the importance of continued efforts in implementing annual MDA. Additionally, if feasible, the use of drug combinations such as tablets containing albendazole and ivermectin may prove beneficial in effectively managing *T. trichiura*, the most prevalent helminth in the region. The study also identified a correlation between SAC attending specific schools and risk of STH infection. To alleviate the burden of STH, we recommend implementing health education and enhancing water, sanitation, and hygiene practices holistically both in schools and associated catchment areas that can act as STH reservoirs. Even though the study did not analyze environmental based factors, environmental based interventions has been shown to reduce the burden of STH.

## Supporting information

**S1 Table. Multivariable analysis of risk factors associated with *A. lumbricoides* infections.**
(DOCX)

**S2 Table. Multivariable analysis of risk factors associated with *T. trichiura* infections.**
(DOCX)

## Acknowledgments

Many thanks to all the ESACIPAC staff, both technical and tertiary for their tireless support. We wish to express our sincere thanks to the Sub-County administrators in Transmara West. The authors also wish to sincerely thank the parents/ guardians who participated in the study. Lastly, we want to express our sincere gratitude to the staff of Kilgoris hospital where the samples were examined for their support. The Director General of KEMRI is also thanked for authorizing the publication of the study results.

## Author Contributions

**Conceptualization:** Richelle W. Kihoro.

**Data curation:** Collins Okoyo.

**Formal analysis:** Collins Okoyo.

**Investigation:** Damaris Mulewa, Dominic Ayaa, Agnes Korir.

**Methodology:** Richelle W. Kihoro, Damaris Mulewa, Collins Okoyo.

**Supervision:** Doris W. Njomo, Charles Mwandawiro, Janet Masaku.

**Writing – original draft:** Richelle W. Kihoro.

**Writing – review & editing:** Richelle W. Kihoro, Damaris Mulewa, Collins Okoyo, Dominic Ayaa, Agnes Korir, Doris W. Njomo, Charles Mwandawiro, Janet Masaku.

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
