## [Decision Letter · Decision Letter 0]

14 Feb 2024

PONE-D-23-39548Epidemiology of soil -transmitted helminthiasis infection among school aged children in pastoralist communities of Narok County, Kenya: A cross-sectional studyPLOS ONE

Dear Dr. Kihoro,

Thank you for submitting your manuscript to PLOS ONE. After careful consideration, we feel that it has merit but does not fully meet PLOS ONE’s publication criteria as it currently stands. Therefore, we invite you to submit a revised version of the manuscript that addresses the points raised during the review process.

We look forward to receiving your revised manuscript.

Kind regards,

Emmanuel Timmy Donkoh, PhD

Academic Editor

PLOS ONE

Journal Requirements:

2. We note that your Data Availability Statement is currently as follows: "All relevant data are within the manuscript and its Supporting Information files."

Reviewers' comments:

Reviewer's Responses to Questions

**Comments to the Author**

1. Is the manuscript technically sound, and do the data support the conclusions?

Reviewer #1: Yes

Reviewer #2: Yes

Reviewer #3: Yes

2. Has the statistical analysis been performed appropriately and rigorously? 

Reviewer #1: Yes

Reviewer #2: Yes

Reviewer #3: Yes

3. Have the authors made all data underlying the findings in their manuscript fully available?

Reviewer #1: Yes

Reviewer #2: Yes

Reviewer #3: Yes

4. Is the manuscript presented in an intelligible fashion and written in standard English?

Reviewer #1: Yes

Reviewer #2: Yes

Reviewer #3: Yes

5. Review Comments to the Author

Reviewer #1: The experiment is well conducted and the article is clearly written and in my opinion merit publication but I would like to request an additional analysis of the data.

MAIN COMMENT

1- ADDITIONAL ANALYSIS

Since the authors collected with Kato-Katz the intensity of infection, I think it would be essential also provide the prevalence of the 4 classes of intensity according WHO : (negative- light intensity- moderate intensity- heavy intensity-) , this classification would allow to better understand if after all the rounds of MDA provided in the last years, the morbidity due to STH is significantly present or not and where: if the in the different schools, infections of modrate/heavy intensity will be less than 2%, we can interpret the data collected aknowleging that the primary objective of MDA has been reached.

This analysis of the data is much more informative in my opinion than the mean intensity and I suggest to completely revise the chapter from line 213 to 242 and table 2 with a chapter and table reporting the prevalence of the 4 WHO classes of intensity.

Also the discussion should be revised according the fact if STH morbidity is still present or not.

I am sure that this can be easily done by the authors and this will provide more inside on the impact of MDA.

Minor Issues

2- STUDY SITE I would provide more information, if available, on the number of rounds of MDA provided to the children in this area during last years, and the reported coverage ( low coverage could be one of the reason for insufficient impact of MDA)

3- ETHICAL ASPECT The authors should mention if the children resulting positive in the survey have been treated and how

4- RESULTS OF MULTIVARIATE ANALYSIS : since the school seems to be the most important factor for infection prevalence, the authors should better elaborate about the possibility that some of the school reached low MDA coverage (for example if the schools are more difficult to access or there reduced school attendance or again there is less compliance among the teachers or children)

Reviewer #2: This is well a written manuscript and has a potential of improving both the local and global interventions towards control of the STHs. It is also rather explicit, easy to understand and straightforward. The data is well presented, neatly arranged and in line with the conclusions.

Comments

Line 23 – add County after word Narok

Line 31 – 37 I find it encouraging to find that the data indicate hookworm prevalence is zero. Did the study seek or want to share the possible causes of the success stories from the ground to be emulated to control the other STH infections eg Trichuris trichiura?

This area of study is known to have a close interface between human, livestock and wildlife animals. It would probably be good to enrich discussion on this sphere. This perspective is important probably while designing control strategy

Line 39 – 44 The conclusion that there should be improvement of the WASH in selected schools is limiting. I suppose that the infections could not only be attributable to the schools alone. It could be possible that there is lapse of sanitation in homes/associated environment and thus acting as disease reservoir(s). The recommendation in my opinion would be to improve WASH holistically both in the school and the associated catchment area although the areas may not have been investigated in the current study.  

Line 81 – I suggest replacement of the word country with Kenya so that it can be more specific.

Line 97 – First word should be Dominant not dominate.

Line 145 - Parasitological examination

It would be good to provide details of: - what power was used for microscopy examination, how the values were translated to eggs per grams and reference the classifications of egg intensity if it was of interest in the study. Was there random examination of the slides for QC/QA especially for the purposes of infection intensity accuracy?

Study design

Although of sampling schools was purposive, how did the study ensure minimal bias and put consideration of equal representation of areas? This would enable all areas to be represented and prevent over sampling of the possible disease hotspots?

Results

Line 174 – Why was this school - Olookwaya 86 (16.7%) under sampled in terms of participants compared to the rest of the schools. Interestingly, the school had the highest prevalence of Trichuris trichiura which also comprises the largest burden of the specific STH in the study.  Is it by design or an oversight?

Line 213 - Mean intensity of the infections

Would it be possible to express this in reference to WHO classifications?

Conclusions

Line 383 ¬– the conclusion proposes use of drug combination that are effective for use to control infection but ran short of mentioning the recommended components or what should be included/added to the current therapy? Any leads? Make the recommendation more specific!

According to this study, is there need for incorporating environmental based interventions as a complementary strategy? 

Reviewer #3: The paper is well written, very clear and and no typos well done. 

I presume that there is more current data other than the year five results that are presented in the paper- It would not be true to present data from 2018 as the current data. There is a paper from Okoyo et al. 2023 that provides lower prevalence for Narok maybe just to acknowledge that since the data that is presented there has been further work in Narok. Have there been any more MDAs in the area?

How were sites selected did this follow the survey design for WHO to conduct surveys to be able to make recommendations, reading the manuscript maybe the title should talk about Trans Mara rather than Narok as all the schools included are from Trans Mara, how generalizable is this data to the rest of Narok

What is the rationale for how the age is divided?

The table 3 can be made smarter to only include known risk factors of interest, why did the analysis include positive children? how is the risk tested for just positive the n values are a bit confusing

6. PLOS authors have the option to publish the peer review history of their article (what does this mean?). If published, this will include your full peer review and any attached files.

Reviewer #1: **Yes: **Antonio MONTRESOR

Reviewer #2: No

Reviewer #3: No

---

## [Author Response · Author response to Decision Letter 0]

17 Apr 2024

Dear Dr. Donkoh,

Re: Revised version of manuscript reference PONE-D-23-39548

Thank you for the opportunity to revise our manuscript titled " Epidemiology of soil -transmitted helminthiasis infection among school aged children in pastoralist communities of Trans Mara West Sub-County, Narok County, Kenya: A cross-sectional study " for submission to PLOS ONE. We are grateful for the valuable feedback from both you and the reviewers, which has helped enhance our paper. We are pleased to resubmit the article for your further consideration, incorporating changes based on your insightful suggestions. We believe our revisions and the responses provided below adequately address the issues and concerns raised

For your review here is a point by point response to the comments and questions raised. The comments are indicated in bold while our responses are not bold.

Journal Requirements

We have gone through the PLOS ONE style templates attached and edited the document accordingly to meet all the requirements.

2. Comment on the Data Availability Statement 

The data availability statement has been edited accordingly. The data set used for this manuscript is part of a bigger project that is currently ongoing. Access to the anonymized and de-identified data set can be obtained by emailing the Kenya Medical Research Institute (KEMRI) Scientific Ethics Review Unit (SERU) at seru@kemri.go.ke this can ensure long term stability and availability of the data is on Line 468 to 469

Reviewer 1 Comments

1. Additional Analysis required for prevalence of the 4 classes of intensity according WHO and adjustment of the discussion according to the fact if STH morbidity is still present or not

Agreed, classifying the prevalence of the four classes of STH intensity according to WHO provides useful information. We have re-analyzed the data according to this comment and provided a section in the results section titled “Prevalence of light, moderate and heavy intensity of infections”, this information can be found in Line 264 to Line 287. Additionally, we have added a paragraph in the discussion section explaining the STH morbidity levels in our study area and possible explanations for this (Line 373 to Line 379).

2. Study Site request for additional information on the number of rounds of MDA and coverage

Thank you for this suggestion we have added a table below the study design and site section (Table 1) that elaborates on the STHs MDA rounds and treatment coverage over time (Line 128 to Line 131).

3. Ethical Aspect on treatment of positive children 

You have raised an important concern, children who tested positive received albendazole (400mg) treatment for STHs following the guidelines of the MoH and the WHO, this has been explained in Line 167 to 169

4. Results of Multivariate Analysis: since the school seems to be the most important factor for infection prevalence, the authors should better elaborate about the possibility that some of the school reached low MDA coverage (for example if the schools are more difficult to access or there reduced school attendance or again there is less compliance among the teachers or children

Thank you for this comment. We have summarized the treatment coverage in Table 1 in the manuscript (Line 128 to Line 131).

Reviewer # 2 Comments

1. Line 23 – add County after word Narok

Agreed, this has been added in Line 30

2. Line 31 – 37 I find it encouraging to find that the data indicate hookworm prevalence is zero. Did the study seek or want to share the possible causes of the success stories from the ground to be emulated to control the other STH infections eg Trichuris trichiura?

Indeed, it is encouraging that they were no hookworm infections. We have attempted to share possible causes of this though the mode of transmission of hookworm and T. trichiura is different and control measures vary. This can be summarized as majority of the participants wore shoes potentially reducing hookworm infections, suboptimal conditions hindered hookworm life cycle, hookworms have been found to be more prevalent in adults than children and higher temperatures may have led to the increased prevalence of T. trichiura and A. lumbricoides while reducing that of hookworms. These points have been expounded in the discussion section from Line 390 to Line 406. 

Additionally, T. trichiura has been shown to have reduced sensitivity to albendazole which is the current medication employed for MDA leading to its high prevalence compared to hookworms that respond well to MDA, this has been explained in Line 390 to 392. 

3. This area of study is known to have a close interface between human, livestock and wildlife animals. It would probably be good to enrich discussion on this sphere. This perspective is important probably while designing control strategy

Thank you for this important insight indeed Narok County has a close interface between human, livestock and wildlife animals. We have enriched the discussion in this respect with an additional paragraph citing a study which detected zoonotic hookworms in dog faces in Narok County and a study that also detected all four known species of canine hookworms in Morogoro, Tanzania. We have also made recommendations on appropriate control measures. All this information can be found in the discussion paragraph from Line 407 to 412. 

4. Line 39 – 44 The conclusion that there should be improvement of the WASH in selected schools is limiting. I suppose that the infections could not only be attributable to the schools alone. It could be possible that there is lapse of sanitation in homes/associated environment and thus acting as disease reservoir(s). The recommendation in my opinion would be to improve WASH holistically both in the school and the associated catchment area although the areas may not have been investigated in the current study. 

Thank you for suggesting this we have amended the conclusion sections (Line 46 to 51) in the abstract and in the rest of the document (Line 445 to 453) to capture a holistic WASH recommendation to all possible areas that may act as reservoirs to alleviate the burden of STH

5. Line 81 – I suggest replacement of the word country with Kenya so that it can be more specific.

Agreed, this has been changed in Line 89

6. Line 97 – First word should be Dominant not dominate.

Agreed, this has been changed in Line 105

7. Line 145 - Parasitological examination It would be good to provide details of: - what power was used for microscopy examination, how the values were translated to eggs per grams and reference the classifications of egg intensity if it was of interest in the study. Was there random examination of the slides for QC/QA especially for the purposes of infection intensity accuracy?

Thank you for raising important points here. The parasitological examination section in the manuscript has been revised to capture the power of microcopy used for examination which was x100 and a description of how eggs per grams was calculated by multiplying the egg counts with a standard conversion of 24. Additionally, a random examination of 10% of the slides were re-read by senior technologists for quality assurance. This information can be found on Line 160 to 170 in the manuscript. We have also re-analyzed the data and classified infection intensities into light, moderate and heavy infections according to WHO guidelines and added a section in the results titled “Prevalence of light, moderate and heavy intensity of infections”, this information can be found in Line 264 to Line 287.

8. Study design- Although of sampling schools was purposive, how did the study ensure minimal bias and put consideration of equal representation of areas? This would enable all areas to be represented and prevent over sampling of the possible disease hotspots?

Thank you for this comment. Sampling was done purposively based on schools within the NSBDP that had a high prevalence of STH (Line 136 to 138). Random sampling techniques were not employed to minimize bias. 

9. Results- Line 174 – Why was this school - Olookwaya 86 (16.7%) under sampled in terms of participants compared to the rest of the schools. Interestingly, the school had the highest prevalence of Trichuris trichiura which also comprises the largest burden of the specific STH in the study. Is it by design or an oversight?

Thank you for this comment. A sample size of 100 children per school was required to conduct the study but some children in Olookwaya refused to participate in the study but in terms of data analysis the study is not overpowered and the overall power for the school was not interfered with. 

10. Line 213 - Mean intensity of the infections- Would it be possible to express this in reference to WHO classifications?

Agreed, classifying the prevalence of the four classes of STH intensity according to WHO provides useful information. We have re-analyzed the data according to this comment and provided a section in the results section titled “Prevalence of light, moderate and heavy intensity of infections”, this information can be found in Line 264 to Line 287. Additionally, we have added a paragraph in the discussion section explaining the STH morbidity levels in our study area and possible explanations for this (Line 373 to Line 379).

11. Conclusions- Line 383 - the conclusion proposes use of drug combination that are effective for use to control infection but ran short of mentioning the recommended components or what should be included/added to the current therapy? Any leads? Make the recommendation more specific!

Thank you for this suggestion this we have indicated that ivermectin should be added to the current therapy used in Kenya as it has shown to be beneficial in managing T. trichiura (Line 448 to 450)

12. According to this study, is there need for incorporating environmental based interventions as a complementary strategy?

Thank you for this suggestion incorporating environmental based intervention is a good recommendation (Line 454 to 456)

Reviewer # 3 Comments

1. I presume that there is more current data other than the year five results that are presented in the paper- It would not be true to present data from 2018 as the current data. There is a paper from Okoyo et al. 2023 that provides lower prevalence for Narok maybe just to acknowledge that since the data that is presented there has been further work in Narok. Have there been any more MDAs in the area? 

Thank you for this comment. We have amended the introduction section to include year 10 results that are reported by Okoyo et al. 2023. This paper has also been cited in the discussion section that shows that Narok county since the commencement of the NSBDP has always had an overall prevalence exceeding 20% for any STH infection (Line 358 to 363). We have also added a table below the study design and site section (Table 1) that elaborates on the STHs MDA rounds and treatment coverage over time (Line 128 to Line 131).

2. How were sites selected did this follow the survey design for WHO to conduct surveys to be able to make recommendations, reading the manuscript maybe the title should talk about Trans Mara rather than Narok as all the schools included are from Trans Mara, how generalizable is this data to the rest of Narok. 

Thank you for this comment. The selection of sites followed the established survey design outlined by the World Health Organization (WHO). According to WHO recommendations, our survey methodology involved the inclusion of 50 children per school or village, with a minimum of five villages representing each ecological zone (World Health Organization, 2018). In adherence to this guideline, a sample size of 100 pupils per school was utilized in our study (refer to Line 140 to 146 in the manuscript).

To address the concern regarding the title, it has been revised to specify the sub-county surveyed, which is Trans Mara. This adjustment ensures clarity and accurately reflects the geographical focus of our study.

3. What is the rationale for how the age is divided? 

The age categories were arbitrarily chosen into three categories to give sufficient numbers for data analysis 

4. The table 3 can be made smarter to only include known risk factors of interest, why did the analysis include positive children? how is the risk tested for just positive the n values are a bit confusing

Thank you for this comment. Risk factor analysis was conducted among a sub-set of the children (139) who were administered the questionnaire. Univariable analysis was conducted to determine association between infection prevalence and socio-demographics, household and knowledge of STH factors reporting odds ratios (ORs) and the 95%CIs while taking into account clustering by schools, n indicates the number of positive children and risk factor analysis was conducted to determine the association of the factor predicting the ability of a positive value. 

Authors wish the version will satisfy the editor and reviewers and meet all standards of PLOS ONE. The authors welcome further comments, if any.

Yours sincerely,

Richelle W. Kihoro

Corresponding author

---

## [Decision Letter · Decision Letter 1]

30 Apr 2024

PONE-D-23-39548R1Epidemiology of soil- transmitted helminthiasis infection among school aged children in pastoralist communities of Trans Mara West Sub-County, Narok County, Kenya : A cross-sectional studyPLOS ONE

Dear Dr. Kihoro,

Thank you for submitting your manuscript to PLOS ONE. After careful consideration, we feel that it has merit but does not fully meet PLOS ONE’s publication criteria as it currently stands. Therefore, we invite you to submit a revised version of the manuscript that addresses the points raised during the review process. The authors are commended for the effort made to revise the manuscript in line with comments raised. However, before the work can be progressed, the Title will require some attention to comply with Journal requirements for clarity and conciseness. See suggestions below. There is also an outstanding comment from Reviewer 1 that requires attention. 

We look forward to receiving your revised manuscript.

Kind regards,

Emmanuel Timmy Donkoh, PhD

Academic Editor

PLOS ONE

Journal Requirements:

Additional Editor Comments:

Authors should note the journal's requirement to make titles specific, descriptive, concise, and comprehensible. The title of the current work should be revised to omit geographical references to read, concisely: "Epidemiology of soil-transmitted helminthiasis among school-aged children in pastoralist communities of Kenya : A cross-sectional study." Also "helminthisis" already connotes infection. Also, the proper punctuation is "school-aged" not "school age."

There is also an outstanding comment from Reviewer 1 that requires attention.

Reviewers' comments:

Reviewer's Responses to Questions

**Comments to the Author**

1. If the authors have adequately addressed your comments raised in a previous round of review and you feel that this manuscript is now acceptable for publication, you may indicate that here to bypass the “Comments to the Author” section, enter your conflict of interest statement in the “Confidential to Editor” section, and submit your "Accept" recommendation.

Reviewer #1: All comments have been addressed

Reviewer #2: All comments have been addressed

2. Is the manuscript technically sound, and do the data support the conclusions?

Reviewer #1: Yes

Reviewer #2: Yes

3. Has the statistical analysis been performed appropriately and rigorously? 

Reviewer #1: Yes

Reviewer #2: Yes

4. Have the authors made all data underlying the findings in their manuscript fully available?

Reviewer #1: Yes

Reviewer #2: Yes

5. Is the manuscript presented in an intelligible fashion and written in standard English?

Reviewer #1: Yes

Reviewer #2: Yes

6. Review Comments to the Author

Reviewer #1: The authors satisfactory responded to all the main points raised by my review and the article in my opinion can be published.

However, I think the authors should do a minor modification to the text:

The point I raised as #4 in my initial review has not been addressed by the authors.

“4. since the school seems to be the most important factor for infection prevalence, the authors should better elaborate about the possibility that some of the school reached low MDA coverage (for example if the schools are more difficult to access or there reduced school attendance or again there is less compliance among the teachers or children”

I understand the coverage data are not available at school level and therefore it is not possible to assess if the differences in STH prevalence measured at school level can be due to difference in coverage. I suggest however to mention this in the limitation chapter of the manuscript.

Reviewer #2: Having relooked at the resubmitted manuscript alongside the original submitted copy and reviewer comments, i feel that the author has done thorough corrections/alignment including further analysis. The manuscript looks good and i therefore recommend for its publication.

7. PLOS authors have the option to publish the peer review history of their article (what does this mean?). If published, this will include your full peer review and any attached files.

Reviewer #1: No

Reviewer #2: **Yes: **Henry Kanyi

---

## [Author Response · Author response to Decision Letter 1]

6 May 2024

Re: Revised version of manuscript reference PONE-D-23-39548

Thank you for the opportunity to revise our manuscript titled "Epidemiology of soil-transmitted helminthiasis among school-aged children in pastoralist communities of Kenya: A cross-sectional study" for submission to PLOS ONE. We are grateful for the valuable feedback from both you and the reviewers, which has helped enhance our paper. We are pleased to resubmit the article for your further consideration, incorporating changes based on your insightful suggestions. We believe our revisions and the responses provided below adequately address the issues and concerns raised

For your review here is a point by point response to the comments and questions raised. The comments are indicated in bold while our responses are not bold.

Journal Requirements

1. Please review your reference list to ensure that it is complete and correct 

We have gone through the reference list and have ensured that all the cited papers are complete and correct and no cited paper has been retracted. From the initial submission I added five references which have been listed appropriately as reference number 29, 38, 39, 41 and 42 in the reference list. The complete reference list can be found from Line 490 to 663. 

Additional Editor Comments 

1. Comment on the title to be specific, descriptive, concise and comprehensible as per journal requirements

Thank you for this comment. I have revised the title accordingly to reflect the journal requirements as "Epidemiology of soil-transmitted helminthiasis among school-aged children in pastoralist communities of Kenya: A cross-sectional study" 

Outstanding Reviewer 1 Comment

1. The point I raised as #4 in my initial review has not been addressed by the authors.

“4. since the school seems to be the most important factor for infection prevalence, the authors should better elaborate about the possibility that some of the school reached low MDA coverage (for example if the schools are more difficult to access or there reduced school attendance or again there is less compliance among the teachers or children” I understand the coverage data are not available at school level and therefore it is not possible to assess if the differences in STH prevalence measured at school level can be due to difference in coverage. I suggest however to mention this in the limitation chapter of the manuscript.

Thank you for this comment, indeed it is a limitation of the study as MDA coverage was reported at the sub-county level rather than the school level, due to data availability constraints. Thus, some schools may have experienced low MDA coverage due to factors such as accessibility, reduced school attendance or compliance among teachers or children. This has been explained in Line 443 to 447 of the limitations section of manuscript

Authors wish the version will satisfy the editor and reviewers and meet all standards of PLOS ONE. The authors welcome further comments, if any.

Yours sincerely,

Richelle W. Kihoro

Corresponding author

---

## [Editor Report · Decision Letter 2]

9 May 2024

Epidemiology of soil-transmitted helminthiasis among school-aged children in pastoralist communities of Kenya : A cross-sectional study

PONE-D-23-39548R2

Dear Dr. Kihoro,

We’re pleased to inform you that your manuscript has been judged scientifically suitable for publication and will be formally accepted for publication once it meets all outstanding technical requirements.

Kind regards,

Emmanuel Timmy Donkoh, PhD

Academic Editor

PLOS ONE

---

## [Editor Report · Acceptance letter]

13 May 2024

PONE-D-23-39548R2 

PLOS ONE

Dear Dr. Kihoro, 

I'm pleased to inform you that your manuscript has been deemed suitable for publication in PLOS ONE. Congratulations! Your manuscript is now being handed over to our production team.

Kind regards, 

on behalf of

Dr. Emmanuel Timmy Donkoh 

Academic Editor

PLOS ONE